# Dietary Interventions in Cancer Treatment and Response: A Comprehensive Review

**DOI:** 10.3390/cancers14205149

**Published:** 2022-10-20

**Authors:** Benjamin D. Mercier, Eemon Tizpa, Errol J. Philip, Qianhua Feng, Ziyi Huang, Reeny M. Thomas, Sumanta K. Pal, Tanya B. Dorff, Yun R. Li

**Affiliations:** 1Department of Radiation Oncology, City of Hope Comprehensive Cancer Center, 1500 E Duarte Rd, Duarte, CA 91010, USA; 2Department of Medical Oncology, City of Hope Comprehensive Cancer Center, 1500 E Duarte Rd, Duarte, CA 91010, USA; 3Division of Quantitative Medicine & Systems Biology, Translational Genomics Research Institute, 445 N. Fifth Street, Phoenix, AZ 85004, USA

**Keywords:** cancer, chemotherapy, radiation therapy, radiotherapy, caloric restriction, fasting, ketogenic diet, cancer treatment toxicity, quality of life, dietary intervention, time-restricted diet, intermittent fasting

## Abstract

**Simple Summary:**

Chemotherapy and radiotherapy are essential components to the management of most solid malignancies. These modalities exert their effects primarily by mediating the DNA damage of malignant cells; however, healthy cells are also damaged by the same mechanisms and can incur acute and late side effects resulting in both morbidity and mortality. Dietary interventions have been shown to reduce cancer growth, progression, and metastasis in many different solid tumor models and they show promise for improving cancer outcomes in early phase clinical studies. Here, we review preclinical and clinical studies that examine how dietary interventions can impact cancer treatment toxicity and efficacy in patients who were undergoing chemotherapy and/or radiotherapy. This information can help clinicians tailor the dietary regimens to patients based on their treatment methods and promote larger clinical trials to test the dietary effects on cancer treatment safety and efficacy.

**Abstract:**

Chemotherapy and radiotherapy are first-line treatments in the management of advanced solid tumors. Whereas these treatments are directed at eliminating cancer cells, they cause significant adverse effects that can be detrimental to a patient’s quality of life and even life-threatening. Diet is a modifiable risk factor that has been shown to affect cancer risk, recurrence, and treatment toxicity, but little information is known how diet interacts with cancer treatment modalities. Although dietary interventions, such as intermittent fasting and ketogenic diets, have shown promise in pre-clinical studies by reducing the toxicity and increasing the efficacy of chemotherapeutics, there remains a limited number of clinical studies in this space. This review surveys the impact of dietary interventions (caloric restriction, intermittent and short-term fasting, and ketogenic diet) on cancer treatment outcomes in both pre-clinical and clinical studies. Early studies support a complementary role for these dietary interventions in improving patient quality of life across multiple cancer types by reducing toxicity and perhaps a benefit in treatment efficacy. Larger, phase III, randomized clinical trials are ultimately necessary to evaluate the efficacy of these dietary interventions in improving oncologic or quality of life outcomes for patients that are undergoing chemotherapy or radiotherapy.

## 1. Introduction

Chemotherapy (CT) and radiation therapy, also known as radiotherapy (RT), are commonly used in the treatment of patients with locally advanced or metastatic solid tumors. CT has its origins in the early twentieth century with the first primitive CT drugs consisting of early aniline dyes and alkylating agents. Both of these were developed by German chemist Paul Ehrlich, who himself coined the term “chemotherapy” [1]. Despite the advent of CT, RT and surgical interventions remained at the forefront of cancer care during the twentieth century, until the development of modern CT drugs that were less toxic and more efficacious. CT drugs saw a resurgence in their use at the beginning of the 1960s, with the development of the L1210 leukemia system for treating acute leukemia, with 25% of the cases among children going into complete remission [1]. This success propelled CT back into the spotlight as a viable avenue of research and treatment. With the rise of adjuvant CT regimens in the 1970s, CT was established as a feasible method of cancer treatment with extremely beneficial results [1,2].

CT inhibits the proliferation of cancer cells in many cases by inducing DNA damage, but in other cases by interfering with the cytoskeleton (microtubules, etc.) to inhibit cell division, either causing considerable oxidative stress that is induced via the effects on the mitochondrial function [3]. In many typical CT regimens, such as the administration of platinum agents such as cisplatin, the damage that is caused to DNA by oxidative stress is often mimicked through the formation of intra- and inter-strand DNA cross-links, thereby forming DNA adducts that prevent the DNA replication that is similar to the impact of ROS-mediated DNA damage [4].

Radiation therapy, or radiotherapy (RT), has been a mainstay in the treatment of most cancers for over a century, with nearly 50% of all of the cancer patients receiving some form of RT through the duration of their illness [5]. In some cases, RT is given in tandem or sequentially in combination with various types of CT agents, with the goal of creating a synergistic effect [6]. Whereas these regimens can be effective in the eradication of cancerous tumors, adverse effects are well documented, and these can limit the treatment duration and cause significant discomfort for patients [7].

While RT has technologically advanced dramatically in the last two decades, photon or heavy ion-based RT inevitably damages some of the normal tissue near the tumor [8]. These can be classified as acute, consequential, or late effects. Acute effects occur within the days and weeks of the initial treatment, and they often involve intermitotic cells such as the skin and mucosa. These effects are often the source of considerable suffering for patients. The consequential effects of RT have similar symptoms as the acute effects do, but they emerge when the acute effects are not properly treated or mitigated. Finally, late effects result from the loss of the function of the postmitotic cells, such as those of the brain, spinal cord, kidney, and vascular structures, and typically, they can emerge up to years after the initial exposure [9].

One of the main mechanisms by which RT kills cancer cells is by generating an increase in the local oxidative stress. In the context of cancer RT and CT treatments, this aspect of oxidative stress can directly take advantage of the Warburg effect, a mechanism by which cancerous cells experience a metabolic shift to rely upon glycolysis for ATP generation [10,11,12], in contrast to the healthy cells that rely primarily on oxidative phosphorylation for the generation of ATP. The importance of this oxidative stress can be shown when antioxidants are deliberately administered within RT regimens. The primary function of antioxidants is to eliminate the reactive oxygen species (ROS), the main perpetrators of oxidative stress, and some studies have suggested that antioxidants can therefore decrease the efficacy of the radiation treatment [13].

Oxidative stress can be modulated in somatic cells in a limited and controlled manner through the alterations of an organism’s diet [14]. In a study of in vivo and in vitro colon carcinoma models, 48 h cycles of short-term starvation (STS) feeding exerted an anti-Warburg effect [15]. This anti-Warburg effect was accomplished by forcing normally glycolysis-dependent cancer cells to use oxidative phosphorylation to generate ATP. As many cancer cells and cell lines possess malfunctioning mitochondria and electron transport chains, oxidative phosphorylation is often inefficient and partially uncoupled by ATP synthesis. This results in oxidative damage occurring to the cancer cells and possible apoptosis [15]. Recent studies in animal models have demonstrated that metabolic based therapies, such as ketogenic diet, caloric restriction, and intermittent fasting may decrease the overall occurrence of spontaneous primary tumors and slow the growth of already existing primary tumors [16].

Briefly, caloric restriction (CR) can be defined as a net energy deficit in an individual’s consumption of food as measured by their caloric intake. The benefits of CR in the context of cancer prevention and treatment have been documented for well over a century, with Moreschi first describing that the tumors that were transplanted into the underfed mice did not grow as well as those that were transplanted in the mice that were fed ad libitum (AL) in 1909 [17]. This deficit in caloric intake is generally set between 20% and 40% of the energy consumption from an organism’s initial average intake [18,19,20]. A lower caloric threshold of CR must be defined to prevent the occurrence of under-nutrition or malnutrition. Reaching these nutritional states is likely to counteract many of the known cancer prevention benefits that CR is known to provide [20].

Intermittent fasting (IF), or as it is often referred to in the scientific community, the time-restricted diet (TRD), is a form of dietary intervention which sets limits on the amount of time an individual is allowed to intake calories, as opposed to the content or number of calories that they consume. This has been hypothesized to improve caloric expenditure and perhaps importantly, encourage the body’s use of stored fatty acids [21]. Unfortunately, the exact number of hours that is optimal remains controversial, but studies have examined the fasts lasting from as little as 12 h to as many as 24 h. A variant of IF is “short-term starvation” (STS), which can require the individuals to fast over several days. Because STS can be difficult to undertake for most individuals, many studies have instead examined STS, which permits a small amount of regulated caloric intake and is often referred to as fasting-mimicking diets (FMD) [22]. Many studies on IF and STS have been encouraging. On a daily regimen of fasting for 18 h, the induction of fatty acid metabolism to ketones have been linked to improvements in dyslipidemia and blood pressure [21]. Preliminary studies that have been performed on animal models have shown promise in the use of IF to improve the treatment outcomes of cancers in transgenic and transplant mouse models of neuroblastoma, fibrosarcoma, glioma, melanoma, breast, and ovarian cancers [19].

In contrast, a ketogenic diet (KD) severely limits the consumption of carbohydrates to induce ketosis, thereby forcing the body primarily to mobilize stored fatty acids for its energy needs. To induce ketosis, one must consume primarily calories in the form of fats in tandem with an extremely low-carbohydrate diet (usually <20–50 g) that is supplemented with adequate amounts of protein [23]. KD is not a new concept and has been used successfully to treat disorders such as intractable epilepsy [24,25]. However, a true KD is a challenging diet to maintain durably due to the extremely limited dietary intake of carbohydrates that are required for an individual to remain in ketosis. The effects of KD on the prevention and treatment of cancers have been examined by multiple studies, with one recent study reporting that it can reduce inflammation, a well-known contributor to cancer risk [26].

In this review, we will discuss the potential therapeutic benefits of dietary interventions including CR, IF, and KD among cancer patients who were undergoing RT and/or CT (Figure 1). We will clarify the potential benefits of these dietary interventions in both the attenuation of the acute clinical symptoms resulting from CT and RT, as well as the potential increases in the efficacy of these therapies (Table 1). We have also included preclinical in vivo studies that have been performed on murine models as well as in vitro cellular models to explore the relationships of said dietary interventions and metabolism (Table 2). Finally, we will discuss the current limitations and explore the directions for future research.

## 2. Materials and Methods

The literature search was performed in July 2022 with the use of PubMed, Web of Science, and Google Scholar as our databases. We only identified the literature that was written in English and did not pose a limitation on the publication year. From the above databases, we used the advanced search feature to screen for search terms that were found in the titles of the articles. The search terms included a combination of “caloric restriction,” “fasting,” “intermittent fasting,” “short-term starvation,” “dietary interventions,” “quality of life,” and “ketogenic diet” with “chemotherapy,” “radiotherapy,” “radiation therapy,” “cancer treatment toxicity,” and “cancer treatment side effects”.

We included the primary literature such as randomized control studies, cohort studies, and case series that had a sample size greater than ten patients. We also included pre-clinical studies to show the foundational evidence for the human trials. We excluded abstracts and non-peer reviewed articles from our review, as well as articles in languages other than English, editorials, case reports, and case series that had less than ten patients, and articles that were not relevant to our topic.

Each record is retrieved and collected by two different reviewers independently. No automation was utilized, and each source was manually reviewed before being entered into the compiled list. A risk of bias judgment was performed by multiple reviewers cross-checking findings to determine if a given study was fit for use.

Eligible clinical studies include patients that were 18 years or older, undergoing CT, RT, or both, and implemented at least one dietary intervention (caloric restriction, fasting, or ketogenic diet). Most of the human studies were safety or feasibility-based pilot studies. As such, there are insufficient data to perform a systematic meta-analysis. All of the figures were created using Biorender.

## 3. Dietary Interventions

### 3.1. Caloric Restriction and Fasting-Mimicking Diets Assist in Further Attenuating Tumor Growth with CT/RT Treatment Regimens

As noted previously, CR represents the reduction in the gross number of calories that are consumed daily, which is generally to less than 20–30% of the normal caloric intake [52]. This diet has a well-established effect of decreasing the overall presence of ROS among the somatic cells, likely resulting from the activation of the eNOS pathway via SIRT1 upregulation [14]. CR has been associated with tumor growth attenuation and progression, with experiments in rodent models indicating that CR can reduce the vascular density of tumors [41] (Table 2). In a pilot study, Tang et al. examined the effect of short-term caloric restriction (SCR) among patients with diffuse large B cell lymphoma (DLBCL) receiving R-CHOP CT [37] (Table 1). In line with previous studies, the patients who underwent either a 48 h SCR or a normocaloric diet. They utilized a newer technique to approximate the nutritional disease and outcome known as a phase angle, where a phase angle that is greater than 5° shows better health outcomes [53]. In this study, the patients’ phase angles within the SCR group increased from 4.92° to 5.4° and displayed stable prealbumin levels throughout the R-CHOP cycles [37]. These results suggested that SCR did not negatively impact the patient’s nutritional status. Further, the SCR cohort had a significant increase in the post-CT hematological parameters, such as an increased erythrocyte count [37].

CR has also been found to enhance the immunogenic effect of RT through the downregulation of regulatory T-cells [18]. While cellular senescence has a strong correlation with the prevalence of the tumor immune response, CR could potentially help to modulate this. When comparing T cell priming between young and aged animal models, CR was able to mitigate the age-associated effects in CD4+ T cell priming [46]. In addition to these immunotherapeutic effects, cancer cells have been shown to compete with lymphocytes, among other cell types, for glucose [54]. It is possible that the acute immune response that is induced by RT alongside CR could increase the rate of glycolysis in CD8+ T cells, thereby potentially depleting the amount of glucose that is available in the tumor microenvironment.

In one trial using murine models, a 30% CR was correlated with tumor regression when it was compared to the control group that was allowed to eat AL. Those tumors in the CR animals showed greater rates of apoptosis and less proliferation at a cellular level [54] (Table 2). Another study of mice that were preconditioned for 10 days with 30% CR found that the pretreatment was associated with restored hematogenic organs and an improved intestinal architecture after RT [45].

In a phase II trial, Lugtenberg et al. analyzed QoL in breast cancer patients either in a fasting-mimicking diet (4-day plant-based one, low-level amino acid diet) or a normal diet utilizing EORTC-QLQ-C30 and EORTC-QLQ-BR23 questionnaires prior to each CT cycle [22] (Table 1). An analysis revealed no difference in the QoL scores between the fasting-mimicking diet and the normal diet [22]. After performing a per-protocol analysis, those who were adherent to the fasting-mimicking diet showed an increase in their physical, emotional, and cognitive well-being with lower fatigue and nausea scores when they were compared to the normal diet and non-adherent groups [22].

Interestingly, there may also be sex-based differences in the response to CR. In a study utilizing mouse models, CR mitigated the radiation-induced systemic and intestinal inflammation in females, while the male mice showed an improved gut barrier function [45] (Table 2). These differences were also reflected in the composition of their respective gut microbiomes, whereby the males had an enrichment of short-chain fatty acid-producing bacteria. The females, conversely, showed a marked deficiency in the microbes that are associated with the pro-inflammatory pathways. The administration of antibiotics to the 30% CR group of mouse models negated these benefits [45]. It has been previously established that altering the gut microbiome profiles may impact the cancer incidence and treatment response [55]. This finding suggests that future research into the relationship between diet modulation and the effects on RT will need to consider the gut microbiome composition as a potential factor in enhanced the therapeutic efficacy of it or the reduction in the side effects.

Despite the potential benefits of CR, this dietary intervention is difficult to sustain for most patients [56], particularly because many cancer patients are at risk of excessive weight loss or malnutrition. Because there is a high patient attrition to the CR diet, the pharmacologic methods that target the molecular pathways which are induced by CR may be particularly useful alternatives. Caloric restriction mimetics are a class of drug/dietary supplement that can mimic the metabolic and hormonal effects of caloric restriction, without the denial of the access to essential nutrients. One example of this is 2-Deoxy-D-glucose (2DG), a glucose analogue that lacks a hydroxyl group at C2. 2DG can be phosphorylated by hexokinase in the initial step of glycolysis, but it is unable to be metabolized by phospho-hexose isomerase, and thus, it competitively inhibits glucose utilization. This eventually leads to elevated SIRT1 and p-AMPK protein levels [57]. Though there is insufficient evidence in humans, in vitro and in vivo pre-clinical studies have shown that they successfully mimicked the reduction in the progression and growth of tumors, which was observed in the CR studies [57,58].

Finally, CR diets have been shown to be safe and feasible for patients who were undergoing cancer treatments, so long as the patient’s nutrition is monitored and accounted for. Valdemarin et al. conducted a phase I/II study examining a CR fasting-mimicking diet (FMD) among patients who were undergoing cancer treatment including CT, RT, immunotherapy, TKIs, and other biologics [38] (Table 1). With a total of 81 patients who were enrolled, 62 patients adhered to the FMD diet, 31 patients displayed grade 1–2 symptoms, and the patients’ body compositions remained stable throughout the treatment regimen [38]. From these data, they were able to show that the FMD diet was safe and feasible for the cancer patients who were enrolled. Additionally, they showed a significant reduction in the c-peptide, IGF1, IGFBP3, and leptin levels [38]. A reduction in these circulating growth factors including adipokines, cytokines, and chemokines have been shown to enhance CT and targeted drug therapy efficacy in the PI3K-mTOR pathway [59]. Two other biomarkers, adiponectin and IGFBP1, have been shown to be inversely proportional to the cancer prognosis. When analyzing the peptide levels several weeks after the FMD regimen, adiponectin and IGFBP1 remained high, suggesting a persistent benefit of FMD [38]. Further research is needed to understand how FMD can affect the tumor growth over longer periods of time.

Although it is effective in increasing the treatment efficacy, CR has been shown to be much more difficult to implement as a supplementary dietary intervention in assisting in the CT/RT treatment regimens. This is likely due to the demanding toll that these regimens have on cancer patients. It is because of this and the existence of potential alternatives that the benefits of CR during CT/RT do not outweigh the deleterious effects of it. While FMD has shown some promise in potentially ameliorating this, the current research is not sufficient to support large-scale clinical trials.

### 3.2. Intermittent Fasting Is a Safe and Feasible Way of Increasing the Efficacy of CT/RT Treatment as Well as Decreasing Treatment Toxicity

In contrast to the CR, intermittent fasting (IF), or a time-restricted diet (TRD), is a dietary intervention that restricts the duration of the feeding time, and this is followed by periods of prolonged fasting [21]. These periods of fasting can vary from more than 24 h periods (often referred to as short-term starvation or STS) to periods of only 12 or more hours [60]. As early as 1982, the studies of murine models showed that a 24 h fasting/feeding period improved the radioresistance to whole-body irradiation, as the dietary intervention was correlated with increased survival rates [61]. In the studies of in vitro human liver cell lines, HepG2 and HuH6, the results displayed that there was a notably increased radiosensitivity under the nutrient-starved conditions (Table 2). This was attributed to an increase in mTORC1 activity, which directly correlated with an increase in the radiosensitivity [47]. A further study examined the effects of starvation on glioma cells both in vitro, utilizing murine, rat, and human lines, and in vivo using only mouse models [44] (Table 2). All of the subjects tested underwent a 48 h period of STS, mimicking the starvation conditions, thereby causing a significant reduction in the blood glucose and IGF-1 levels. The reductions in glucose in vitro were strongly correlated with an increased sensitization of the tumor tissue to both RT and CT. In vivo, the STS mice had a significantly lower tumor burden and higher rates of survival after the intracranial GL26 glioma implantation when it was performed in adjunction with Temozolomide (TMZ). However, the standard 48 h STS cycle and the higher TMZ dosage were poorly tolerated. Shortening the fasting cycle to 24 h and reducing the TMZ treatments to just one injection per day prevented the occurrence of an apparent weakness, allowed for the mice to maintain adequate bodyweight, and delayed the onset of tumor-induced mortality [44].

One possible explanation for this phenomenon is that fasting, even for short periods of time, can induce an anti-Warburg effect. This is likely mediated by hypoxia-inducible factor-1 (HIF-1), a central regulator of glycolysis, whose activation has been associated with angiogenesis, erythropoiesis, and the modulation of notable enzymes within aerobic glycolysis [62]. Using a murine xenograft model of CT26 carcinoma cells that were subjected to fasting conditions, the researchers showed a decreased ability of the tumor cells to metabolically adapt, thereby resulting in higher rates of oxidative damage and tumor cell apoptosis [15]. Alternate day STS has also been shown to significantly increase the radiosensitivity among the mouse models of mammary tumors. This is likely due to the increased amount of oxidative stress that is placed on the cancer cells, with the consequence of DNA damage, relative to the surrounding non-cancerous cells [63,64].

Alongside the potential benefits during RT, IF has been investigated in combination with CTs for many years. In mice and yeast models, 48 h STS was associated with a reduction in the hematological toxicity that occurs after CT use, increased rate of DNA repair, and an increased treatment efficacy in combating metastatic neuroblastoma cells [39] (Figure 2; Table 2). In a randomized pilot study of dietary modulation among 13 HER2-negative breast cancer patients receiving CT, de Groot and colleagues demonstrated that a 48 h short-term fasting period was a feasible method of reducing bone marrow toxicity, as measured by the lymphocyte count and the CT-induced DNA damage in the healthy cells [30] (Figure 2; Table 1). In this same study, however, no significant difference was observed in the grade I/II and grade III/IV clinical toxicity between the control and the STS patients, which is likely due to the limited power and limited rate of the patients’ adherence to the study [30]. In another randomized control study, Dorff et al. showed safety and feasibility for 24 h, 48 h, and 72 h STS in patients who were undergoing CT, and they also found that in the 48 h and 72 h STS cohorts, there was a decreased rate of DNA damage in the leukocytes when they were compared to the 24 h cohort and baseline [31] (Table 1).

Following de Groot and Dorff’s studies, Bauersfeld et al. conducted a crossover pilot study examining the effects of STS on the quality of life (QoL) of ovarian and breast cancer patients who were undergoing CT [34] (Table 1). QoL is defined as the physical and psychological well-being of the patient, and previous studies have shown that there is an association between QoL and the patient’s prognosis in different cancers [65]. The patients followed either a 60 h fasting protocol (36 h fast prior to the treatment and 24 h fast after the treatment) or they were provided a normocaloric diet for their first three CT cycles before they switched over to the fasting diet for the following three cycles. The patients completed questionaries such as the functional assessment of cancer therapy-general (FACT-G) one and the trial outcome index of the functional assessment of chronic illness therapy (TOI FACIT-F) on to measure the QoL and fatigue scores, respectively. Among the group that initially fasted, significant increases in the FACT-G and TOI FACIT-F scores, as compared to those of the normocaloric cycle, were reported. Despite this, there was no significant difference in the QoL values among those who fasted after the study cross-over [34]. Bauersfeld suggested that fasting may only be protective when it is employed prior to the onset of the symptoms. Despite this discrepancy, a 60 h STS was still associated with an improved QoL and a reduced amount of fatigue as compared to those of the normocaloric diet regimen across both of the cohorts [34]. Like Bauersfeld et al.’s protocol, Riedinger et al. conducted a randomized control study analyzing the QoL scores of women with gynecological cancers who were undergoing CT [60] (Table 1). Here, they randomly assigned the patients to the short-term fasting cohort (24 h prior to treatment and 24 h after treatment) or the control AL cohort. From their results, they found that fasting was well-tolerated, improved the patient QoL over the course of the cycles, and even showed a trend in reduced hospitalizations and improved hematological parameters [60].

Whereas the metabolic consequences of IF and STS on the tumors have been well-documented, the potential alleviation of CT/RT side effects remains to be fully elucidated. One study among a small cohort of 10 patients who were undergoing CT showed that they experienced benefits from the implementation of a fasting regimen. Six of the ten patients reported a reduction in the fatigue, weakness, and gastrointestinal side effects after fasting prior to CT [66] (Table 1). A cross-over pilot study showed that the patients who were undergoing a 96 h fasting regimen had reduced the amount of stomatitis, nausea, and other chemotherapeutic toxicities when they were compared to the normal diet group [49] (Table 1). Interestingly, conflicting data exist on the effect of IF on specific cellular signaling pathways between the different cell lines. A retrospective cohort study showed no improvement of the tyrosine kinase inhibitor (TKI) efficacy in patients with chronic myeloid leukemia (CML) who were implementing an IF diet [27]. However, an in vivo murine study found that IF potentiates the TKI’s activity in mice that were implanted with human hepatocellular carcinoma cells [67] (Table 2). These studies underscore that IF and STS have the potential to reduce the treatment toxicity in patients who were undergoing cancer treatment, yet the mechanism by which these dietary interventions affect the differing types of cancer in the human patients requires further prospective human studies.

### 3.3. Ketogenic Diet Has Been Shown to Maintain Non-Fat Body Mass and Increase CT/RT Treatment Efficacy in Murine Models and Human Cancer Patients

The ketogenic diet (KD) has been the subject of numerous in vitro and in vivo studies, utilizing many different cell lines of various cancers and mouse models. We will, however, discuss both isocaloric KD and KD, which were administered at a caloric deficit. What demarcates KD from the previously mentioned diets is the specificity with which a potential patient must adhere to certain types of food, as well as the induction of a state of ketosis [68].

In line with previously noted diets, the KD-driven ketosis increases the amount of oxidative stress throughout the body via the increased reliance on fatty acid oxidation. This also facilitates a significant drop in the amount of available blood glucose for any glycolysis-reliant cancer cells. The depletion of glucose could especially be vital to the treatment of glycolysis-dependent cancers in the brain. Here, ketone bodies are primarily utilized in a state of glucose depletion, as fats and proteins cannot cross the blood–brain barrier [29]. The proposed anti-Warburg mechanism would allow for the selective increase in the radiosensitivity among the cancer cells when they are compared to the healthy cells [55].

KD has been noted to exhibit complimentary effects when it is implemented in combination with RT and/or CT in both the isocaloric and calorically restrictive forms. In murine models with Mia PaCa-2 human pancreatic cancer xenografts, mice that were fed with a 4:1 KetoCal diet were significantly more likely to survive after 150 days of ongoing RT regimens than their AL contemporaries were, which were receiving the same RT [33] (Table 2). In the clinical trials that were conducted on patients with a histological diagnosis of pancreatic cancer, a combination of RT, CT, and KD were used consecutively (Table 1). Only two patients were able to successfully comply with the intervention; the remaining patients withdrew due to an intolerance to the diet or a dose-limiting toxicity to it. Additionally, many studies have demonstrated the challenge of KD about compliance and tolerability among patients with varying cancer types. Compliance was defined as the patients being able to fully maintain the diet through all of the cycles of CT and/or the entire duration of their RT treatments [69]. In another study that was conducted among 16 cancer patients with advanced metastatic diseases of varying types, no severe side effects of implementing the diet were observed, except for temporary constipation and fatigue (Table 1). However, patients who were experiencing diarrhea before because of their treatment reproved normalized bowel movements shortly after ketosis was induced [28].

The rationale of utilizing KD concurrently with CT or RT rests on the belief that such a diet will sufficiently drop the blood glucose levels to induce ketosis. Previous studies have suggested that many types of cancer cells are not able to utilize these ketone bodies effectively as sources of metabolic fuel, notably neuroblastoma cells, in addition to other brain tumor cells [43]. The presence of a certain type of ketone body, β-HB, has been associated with the inhibition of HDAC activity, contributing to its known protective effects against oxidative stress [70]. However, one study suggested that certain cancers may still be able to function in these low-glucose, high-ketone body environments. In murine models, KD mice with HeLa cell xenografts had an increased measured tumor growth with a decreased survival in the context of a KD diet (Table 2). In contrast, the mice with PANC-1 xenograft tumors appeared to benefit from KD, with them having vastly inhibited tumor growth and increased survivability [50]. This study also suggested that the ketolytic enzyme expression levels may account for whether a tumor cell will be able to metabolize the ketone bodies. In this way, the level at which the ketolytic enzymes were expressed in each line of tumors could determine how a given tumor cell line would respond to the KD conditions during CT and RT. Higher levels of the ketolytic enzyme expression resulted in worsened therapeutic responses and even increased rates of tumor growth.

Few studies have examined KD and its potential to modulate the side effects that are associated with CT and RT. Those that exist tend to be case studies or involve small sample sizes (≤16 patients), and the results of them have been inconsistent regarding any changes in the severity of the treatment-related side effects. For example, De Groot and colleagues [30] found no significant differences in the symptoms in the context of CT among those on a KD when they were compared to the control groups. In contrast, Schmidt et al. reported improved emotional functioning and reduced rates of insomnia among the patients who were on a KD shortly before, during, and after the CT treatments [28] (Table 1). Existing research shows a general improvement in the QoL, and likely additional weight loss that is associated with KD intervention [35]. Selecting patients for whom weight loss is permissible or beneficial is important to reduce the risk of harm of excess weight loss and malnutrition in patients who are experiencing cancer-related cachexia. Unique to KD adjuvant therapies, however, is the fact that this weight loss is almost entirely in fat mass, and no significant changes in mass were observed in the cells that are not associated with fat storage. This maintenance of fat-free mass in patients who were undergoing KD was attributed to an increased intake of protein relative to a high-carbohydrate content diet in the control group [35]. Further investigation into the mechanism of the muscle mass preservation that is seen with KD would be an advisable avenue of future research, as would the implementation of KD in wider clinical trials.

## 4. Discussion

Most of the studies we identified showed some level of support for a potential benefit of CR, IF, and KD as dietary interventions in patients who were undergoing CT and/or RT. However, many limitations remain in these studies, and we urge strong caution before interpreting or attempting to apply these data with regard to clinical practice. These limitations include them having small sample sizes, low rates of patient adherence to diet regimens in some studies, and high attrition rates, which contribute to the limited statistical power of these studies. To our knowledge, very limited research has been performed on humans of a sizable patient population on this subject.

Most of the trials that are reviewed here were pilot studies, meaning that the sample sizes were often much too small to achieve statistical significance and power. As a consequence of the limited sample sizes, some of these studies also included heterogenous cancer types, making their effects difficult to interpret. Despite this, there is promising work examining the possible biomarkers of the treatment efficacy regarding KD and different types of cancer [50]. There also exist many ongoing clinical trials exploring the subject of dietary intervention, with some even controlling for different variants of cancer (Table 3).

Patient adherence is obviously essential to the success of any intervention and the potential to observe its effect on the treatment outcomes, and QoL is inherently dependent on the patient’s ability to carry out the intervention. Lugtenberg et al. and de Groot et al. both found that 80% of the patients were able to adhere to the fasting-mimicking and IF diets, respectively, for the first cycle of CT. However, only around 20% of the patients adhered to these diets throughout all of the CT cycles [30,55]. Better methods of supporting patients and ensuring adherence are necessary to evaluate the clinical efficacy of dietary interventions in these patient cohorts. The involvement of nutritionists can be one method that can help the patients persevere with their diet regimen as well as maintain adequate nutrition [38].

While KD is difficult to maintain for most patients, CR appears to be even more difficult, which may be related to the prolonged caloric deficit in the CR diet. Evaluating the causes for the attrition may be difficult as studies of KD either required a caloric deficit or did not require isocaloric intake. Nevertheless, the KD studies that required a caloric deficit saw higher attrition rates than the studies of isocaloric did. Zorn et al. displayed in their pilot study that there was a high attrition rate because of the difficulty maintaining the diet and treatment [49]. In this case, closer attention must be paid to the recruitment strategies for future fasting studies to reduce the patient attrition rate and focus the studies on the patients who can undertake the challenging intervention. Overall, while the attrition rates were higher in the CR and KD studies, small-scale clinical trials of all three forms of dietary interventions have observed significant differences in the CT and RT treatment efficacy and toxicity, underscoring the need for further studies of these interventions.

Particular attention should be paid to gastrointestinal (GI) cancers as there is conflicting evidence as to whether the patients with GI cancers would derive benefits from these interventions. While these patients are prone to malnutrition and weight loss during their treatment, many are overweight due to obesity being a major risk factor for cancers of the colon, rectum, esophagus, liver, and pancreas [71]. Nguyen et al. examined how nutritional interventions improved the QoL in patients with GI cancer who were undergoing CT in Vietnam [36] (Table 1). Patients with early-stage GI cancers tend to be asymptomatic and thus this disease tends to be detected later with a poorer prognosis [72]. As a result, QoL is an important endpoint that is used to assess the treatment outcomes among those with GI cancers [65]. When GI patients received high-protein and energy-rich diets, they had higher QoL scores when they were compared to those of the control as well as reduced adverse effects from their CT regimen [36].

Several in vivo studies have shown positive outcomes of the IF, KD, and CR diets in mice with different GI cancers [40,42,48] (Table 2). Klement et al. showed, that five out of six patients saw typical tumor regression, as seen in previous murine studies, alongside a preservation of muscle mass [32] (Table 1). In sum, there appears to be potential for using nutritional diets as an adjuvant for reducing the tumor growth and CT side effects; however, more studies are needed to investigate the functionality and underlying mechanisms of the fasting diets. Parameters such as QoL, nutritional health, and treatment outcome when one is determining the utility of these dietary modifications in the cancer treatment should be assessed as well. Cancer patients are often at a much greater risk of malnutrition than the general population is, especially when they are undergoing rigorous CT/RT treatments. According to the ESPEN Practical Guideline for clinical nutrition in cancer, the current evidence strongly advises against dietary provisions that restrict energy intake such as CR or CR/KD in malnourished patients [73].

The promising results that are reviewed herein provide the impetus for the further investigation of IF in the context of cancer in larger randomized clinical trial patients. CR and KD evidently have been the most difficult for cancer patients to comply with as there have been mixed patient compliance outcomes which have been reported. Further investigations are still needed among larger patient populations to better determine the validity of this claim. As a result, these diets might not be the most ideal method for dietary intervention when they are compared to IF. The research that currently exists largely utilized murine and other mammal models, with there being a few pilot studies in humans. Those that have been conducted with human subjects suggest that the largest issue with the regimens of dietary modulation during radio- or CT in humans may be patient compliance as these patients are under high amounts of stress from their treatment.

Another avenue to consider would be the use of pharmacologic mimetics to replace the dietary interventions, such as a CR diet, through the inhibition of different pathways such as glycolysis and fat metabolic processes [57]. These therapeutics would likely significantly improve the rate of patient adherence. Several in vivo studies have shown the potential benefit of the previously mentioned mimetic, 2DG, as an anticancer agent inhibiting the AMPK signaling pathway [55,56]. Other studies have shown that 2DG can be quite cytotoxic, thereby causing cardiotoxicity in mice [74,75]. It is still not fully understood if these mimetics are entirely safe in cancer patients, nor if they cause drug interactions with cancer treatments. Further pre-clinical and clinical studies are necessary to evaluate how well these mimetics induce the beneficial effects of CR, IF, or KD for patients who were undergoing cancer treatments.

Lastly, further work in the delineation of the different forms of dietary interventions and how each could affect different types of cancer should be considered. The existing studies remain heterogenous in the type of cancer, the stage of the disease, and the treatment modalities that have been examined, thus limiting the ability to draw any generalized conclusions.

## 5. Conclusions

As a whole, the studies that are examined in this review show that dietary intervention such as CR, IF, or KD are feasible and have the potential to be effective in combination with standard of care oncologic therapies in improving the patients’ health, QoL, and treatment outcomes. However, further clinical trials using much larger patient populations must be conducted to explore these before they can be safely implemented in broader patient populations. Whereas pharmacologic interventions can be expensive, increase the medication burden, and are less adaptable to a broad variety of cancers and treatments, the dietary interventions are cost-effective and could be accessed by patients regardless of their socioeconomic status. In addition, these interventions allow the patient to participate in their treatment in a way that could significantly impact outcomes and encourages patient engagement and adherence. While we await the publication of more definitive trials examining these interventions, identifying the optimal implementation strategies, identifying the patients who are most likely to benefit from them as well as exploring the ways to improve adherence will be key to advancing future interventions that can improve the patient outcomes in the clinic.

## Figures and Tables

**Figure 1 cancers-14-05149-f001:**
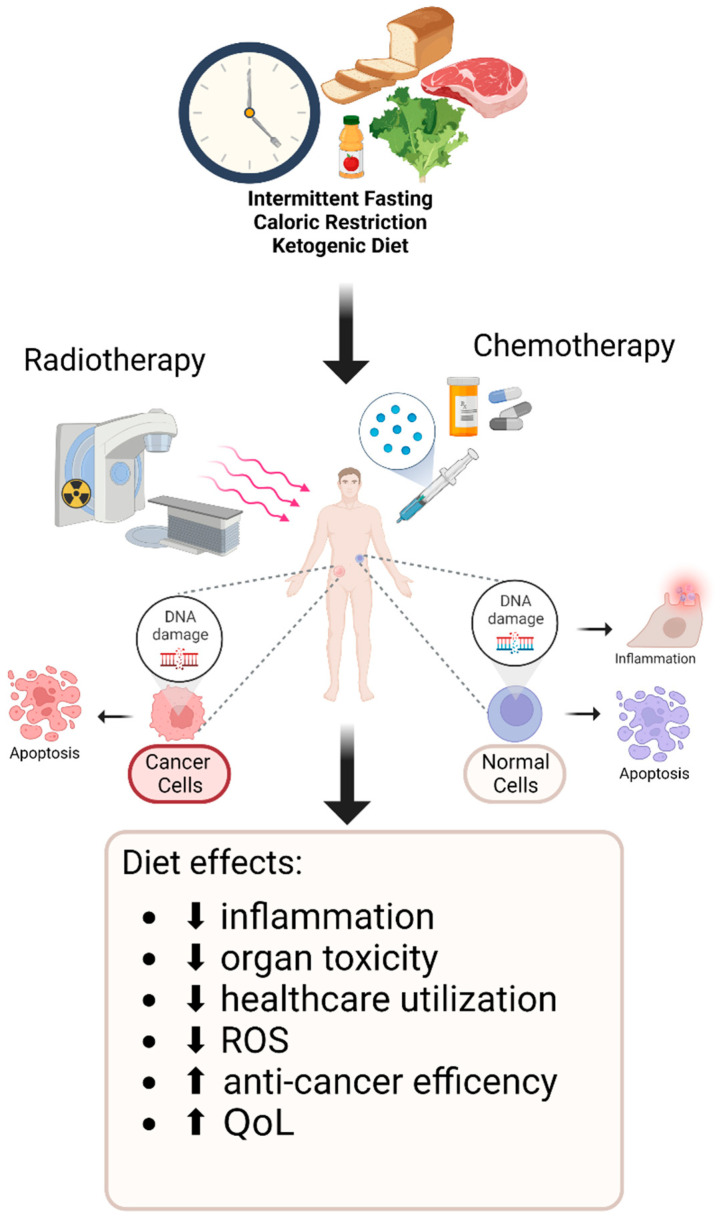
Potential positive effects of dietary interventions on patients receiving cancer treatment. Quality of life (QoL) is a parameter that is used to determine the well-being of an individual both physically and psychologically. Reactive oxygen species (ROS) are unstable molecules which induce DNA damage, thereby leading to cell death.

**Figure 2 cancers-14-05149-f002:**
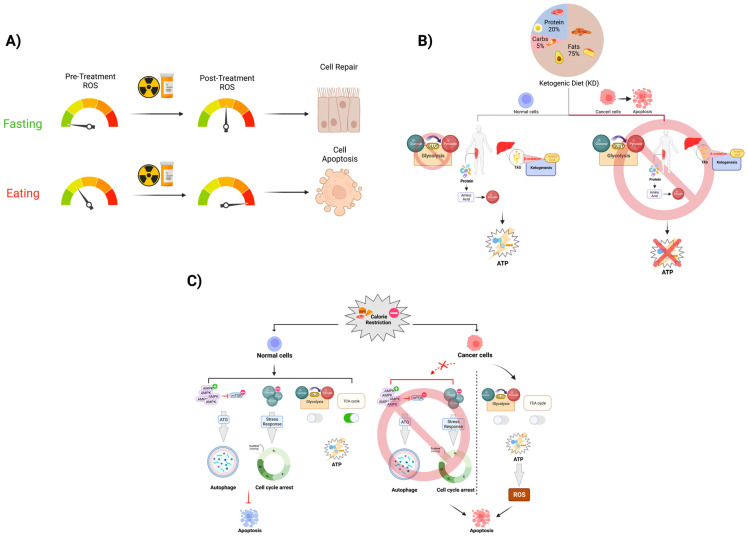
Fasting’s effect on anticancer treatment toxicity. The mechanism of radiation and some chemotherapeutic drugs are largely mediated through the induction of increased levels of oxidative stress-mediated DNA damage in the cell. Most of the oxidative stress in the cell is predominately produced by physiologic metabolic processes. (**A**) During fasting periods in IF, the level of ROS that is produced by the mitochondria is significantly lower than it is during the active metabolism period. By introducing a period of short-term fasting pre- and post-cancer therapy that induces ROS, we expect this dietary modulation to protect normal cells from excess levels of DNA damage due to DNA stress. In this case, more normal cells can repair themselves and survive, while highly metabolic cancer cells are less likely to respond to short-term fasting as they are likely undergoing apoptosis as a result of this. (**B**) As described by the Warburg effect, cancer cells vastly increase their uptake of glucose, developing an overreliance on glycolysis as their primary method of ATP generation. CR utilizes this lack of metabolic flexibility to disproportionately place more oxidative stress on cancer cells than they do on normal somatic cells, thereby leading to apoptosis. (**C**) KD is able to exploit the Warburg Effect as well through purposefully depleting serum glucose via reducing the consumption of carbohydrates. This results in an overall increased reliance on fatty acid beta-oxidation and gluconeogenesis to provide energy and maintain appropriate serum glucose levels, effectively starving cancer cells of glucose and curbing tumor growth rates.

**Table 1 cancers-14-05149-t001:** Clinical studies examining the effect of dietary interventions (CR, IF, and KD) in patients who were undergoing chemotherapy and/or radiotherapy.

Author	Year Published	Years Conducted	Patient Population	Type(s) of Cancer	Treatment Regimen	Diet Regimen	Outcome: QoL	Outcome: Therapy Toxicity/Efficacy
Safdie et al. [27]	2009	April 2008–August 2009	*n =* 10	Breast, esophageal, prostate, lung, ovary, uterus	CT	IF (fasting 48–140 h before)	Six fasting patients reported reduction in fatigue, weakness, and gastrointestinal side effects.	NA
Schmidt et al. [28]	2011	Could not determine	*n =* 16	Breast, esophageal, prostate, lung, ovary, granulosa cell tumor, parotid carcinoma, osteosarcoma (jaw) thyroid carcinoma, colon carcinoma, endometrial	CRT	KD	Five out of sixteen patients completed the study and one dropped out due to diet intolerance. No severe adverse effects were observed, and insomnia improved.	Blood cholesterol was reduced significantly. Blood leukocytes increased significantly.
Champ et al. [29]	2014	August 2010–April 2013	*n =* 53 (6 underwent dieting)	Glioma	CRT	KD	NA	Serum glucose levels were reduced, and diet was safe/feasible.
de Groot et al. [30]	2015	May 2011–December 2012	*n =* 13	Breast	CT	IF (48 h short-term fast)	NA	Mean erythrocyte and thrombocyte counts were higher in the IF group. Non-hematological toxicity did not significantly differ, but γ-H2AX levels were significantly higher in CD45 + CD3- cells in the non-IF group.
Dorff et al. [31]	2016	October 2009–November 2012	*n =* 20	Urothelial, NSCLC, ovarian, uterine, breast	CT	IF (24 h, 48 h, and 72 h STS)	Tolerance to fasting was assessed and limited to ≤grade 2 dizziness, fatigue, and headache. Symptoms tended to increase as fasting period increased.	DNA damage in leukocytes for patients who fasted 48 h or more was reduced. IGF-1 levels decreased by 30%, 33%, and 8% in the 24 h, 48 h, and 72 h cohorts, respectively.
Klement et al. [32]	2016	2014	*n =* 6	Breast, prostate, rectum, primary lung	CRT	KD	QoL was stable and good through CRT, but eventually decreased, possibly from prolonged radiation exposure. Some patients experienced significant weight loss with KD at a caloric deficit.	Tumor regression occurred as expected in 5 of 6 patients, with 1 seeing slight progression of small cell lung cancer. Once KD ended, their disease progressed rapidly.
Zahra et al. [33]	2017	July 2011–June 2014	*n =* 9	Lung, pancreatic	CRT	KD	2 of 7 lung cancer patients and 1 of 2 pancreatic cancer patients maintained KD. All who dropped out noted compliance difficulties.	Compliance was 33%, too difficult to assess outcome.
Bauersfeld et al. [34]	2018	November 2013–August 2015	*n =* 34	Breast, ovarian	CT	IF (started 36 h before and ended 24 h after)	FACT-G and FACIT-F scores increased in the IF group compared to the non-IF group. Fasting QoL scores increased while non-fasting QoL scores decreased. QoL scores in cross-over fasting groups remained similar.	NA
Klement et al. [35]	2019	Could not determine	*n =* 81	Head and neck, rectum, breast	RT	KD (also supplemented with 10 g of essential amino acids on days with RT)	No significant adverse effects reported.	In rectal and breast cancer patients, KD significantly associated with fat mass loss. No significant changes in nonfat and skeletal muscle mass occurred. HNC patients who were under CRT had KD-associated gain in fat-free and skeletal muscle mass
Nguyen et al. [36]	2021	2016–2019	*n =* 120	Gastrointestinal	CT	Nutritional Intervention	Maintenance of adequate nutrition in GI cancer patients saw significantly higher QoL assessment scores compared to control groups.	Nutritional intervention was associated with significantly better treatment prognoses.
Tang et al. [37]	2021	October 2020–July 2021	*n =* 12	Diffuse Large B-cell lymphoma	CT	CR (short term)	Diet was tolerated and compliance was high. Mean phase angle increased compared to the baseline.	CR intervention was associated with higher erythrocyte and lymphocyte counts.
Valdemarin et al. [38]	2021	November 2017–August 2020	*n =* 90	18 different types of cancer, plurality was breast	CRT	CR/IF (fasting-mimicking)	Diet was found safe/feasible. Increase in phase angle and fat-free mass was noted while fat mass decreased.	Serum c peptide, IGF-1, IGFBP3 and leptin levels decreased while IGFBP1 increased.

QoL: quality of life, CT: chemotherapy, IF: intermittent fasting, NA: not assessed/not available, CRT: chemoradiation therapy, KD: ketogenic diet, STS: short-term starvation, GI: gastrointestinal, and CR: caloric restriction.

**Table 2 cancers-14-05149-t002:** Pre-clinical studies examining the effect of dietary interventions (CR, IF, and KD) in animal models undergoing chemotherapy and/or radiotherapy.

Author	Year Published	Animal/Cellular Model	Type(s) of Cancer	Sample Size	Condition Tested	Outcome
Raffaghello et al. [39]	2008	S. cerevisiae with RAS2 and/or SCH9 deletion. Rat cell lines: primary rat mixed glial cells (astrocytes + 5–10% microglia), C6, RG2, A10-85, and 9 L Human cell lines: LN229, SH-SY5Y	Glioma, neuroblastoma	~260 million S. cerevisiae cells	48-h STS in S. cerevisiae; noncontrol cells were treated with either hydrogen peroxide or menadione. MMS and CP were used to mimic chemotherapy treatment.	STS cells with RAS2/SCH9 deletions were up to 1000-fold more resistant to oxidizing treatments. Expression of the oncogene-like RAS2^(Val19) negated this protection.
		Rat cell lines: primary rat mixed glial cells (astrocytes + 5–10% microglia), C6, RG2, A10-85, and 9 L Human cell lines: LN229, SH-SY5Y	Glioma, neuroblastoma	5 different in vitro cell lines	Concentration of glucose in experimental group was reduced to mimic STS, STS induction of DSR in mammalian cell lines, cells were treated with H_2_O_2_, menadione, and CP	80% STS survival rate vs. <10% for control glial cells. No difference between cell survival in H_2_O_2_, but STS induced DSR in normal cells treated with methadione, causing cancer cell sensitization/lower survival rates. No difference was observed in CP treated STS cells.
		A/J mice	NA	*n =* 30, 17 under STS, 23 AL; *n =* 10 for 60 h STS	Mouse models were injected with EtO to generate cytotoxic conditions, 80 mg/kg dose of EtO was administered	43% of control mice died by day 10; only 1 of 17 STS-treated mice died after EtO treatment. STS mice showed no visible signs of stress or pain after EtO treatment; control mice showed reduced mobility and increased stress. No STS mice died in 60 h STS trials.
		A/J mice	Neuroblastoma	*n =* 46	Mouse models were inoculated with NXS2 cells. Survival rates of 48 h STS, non-STS, and STS/EtO administered mice were compared.	NXS2/STS/EtO had a significantly larger survival rate than NXS2/STS or NSX2 mice
Otto et al. [40]	2008	NMRI strain mice	Gastric adenocarcinoma	*n =* 24 (total), *n =* 12 (KD group), *n =* 12 (standard diet)	Mice were injected subcutaneously with cell line 23132/87 and randomly split into KD or standard diet groups.	KD group tumors had significantly delayed growth. After day 20, KD tumor growth increased, yet remained significantly lower. KD group tumors had significantly larger necrotic areas.
Powolny et al. [41]	2008	Copenhagen rats	Prostate adenocarcinoma	*n =* 10 (total), *n =* 5 (AL), *n =* 5 (40% CR)	Rats were fed either AL or 40% CR. After 8 weeks, rats were implanted with AT6.3 cells. 2 weeks after, serum concentrations and protein expression of IDF-I, IGFBP2, and VEGF were measured.	CR reduced IGF-1 (35%) and increased IGBP3 by 7-fold. mRNA expression of IGF-1 and its receptor, as well as VEGF mRNA/protein secretion, were significantly lowered. CR reduced endocrine/autocrine IGF-1 expression, contributing to reduced VEGF/angiogenesis.
Colman et al. [42]	2009	Rhesus macaques	NA	*n =* 76 (total), *n* = 30 (1st cohort), *n =* 46 (2nd cohort)	Rhesus macaques were studied longitudinally over the course of 20 years, with some undergoing moderate CR and compared to a control-fed group.	50% (control) vs. 80% (CR) overall survival
Skinner et al. [43]	2009	Human foreskin fibroblasts (control) and human neuroblastoma cells (SK-N-AS)	Neuroblastoma	1 control and 1 neuroblastoma cell line	2 different cell lines were grown in standard media with or without glucose (glc +/glc-), glc- with acetoacetate, or glc- with BHB. This was to determine cell viability in ketone-rich environments.	Neuroblastoma viability decreased significantly in BHB and acetoacetate media. Inability for neuroblastoma cells to utilize ketone bodies was likely due to decreased expression of the SCOT protein.
Safdie et al. [44]	2012	Primary mouse glia, murine GL26, rat C6, and human U251, LN229, and A172 glioma cells	Neuroblastoma	6 different cell lines: 2 mouse, 1 rat, and 3 human	TMZ was tested in both AL and STS-mimicking conditions	STS-mimicking sensitized glioma cell lines, not glial cells, to TMZ.
		C57BL/6N murine models injected with GL26	Neuroblastoma	*n =* 30 (subcutaneous), *n =* 33 (intracranial)	TMZ administered in both STS and AL conditions on in vivo murine models with GL26 injected both subcutaneously and intracranially	48 h STS caused a significant decrease in blood glucose and IGF-1 levels and significant sensitization to CRT treatment.
Saleh et al. [45]	2013	Balb/c murine models	Triple negative breast (TNBC)	*n =* 80	67NR/4TI cell lines were injected into mice part of either a standard diet group, or an ADF group. Desired CR was not achieved, so a second cohort of mice were injected with 4TI TNBC cells. All groups were subject to RT.	ADF caused tumor growth delays in 67NR and 4TI tumors. CR + RT saw the greatest growth delay. No difference between the first and second ADF cohorts was specified.
Farazi et al. [46]	2014	Balb/c and C57BL/6 murine models	Mouse fibrosarcoma	NA	Aged mice were placed on CR or dietary supplementation with RES, a CR mimetic. Tumor immune responses were induced and observed starting at 4 months until the mice reached ~12 months in age.	CR resulted in fully sustained OX40-mediated anti-tumor immunity and antigen-specific CD4 T cell priming in aged hosts. RES supplementation was unable to mimic such effects.
Murata et al. [47]	2015	HepG2 and HuH6 cell lines; SV-0-transformed human fibroblast line (LM217)	Hepatoma	3 cell lines	Nutrient starvation was induced in all 3 cell lines to observe effects on mTORC1 activity and other cellular signaling pathways.	mTORC1 activity was suppressed in LM217 cells under nutrient starvation. Hep2 and HuH6 cells increased in mTORC1 activity and radiosensitivity. mTOR inhibition by siRNA/rapamycin suppressed this in HepG2.
Bianchi et al. [15]	2015	CT26 colorectal tumor cells, Balb/c murine models	Colorectal	1 cell line, *n =* 28	Cells were subject to 48 h STS or given standard nutrients. In either group, half were given OXP. The same analysis was performed on mouse models, with CT26 injections in their lower back.	In both in vivo and in vitro models, STS was shown to exhibit an anti-Warburg effect, causing tumor cells to move into an uncoupled oxidative phosphorylation model of metabolism.
Sun et al. [48]	2017	CT26 colorectal tumor cells, Balb/c murine models	Colorectal	*n =* 12, *n =* 6(ADF), *n =* 6 (control)	M2 TAMs promote cancer cell proliferation. Experimental in vitro and in vivo models were both placed under fasting conditions and compared to control cell lines/mouse models.	Fasting was shown to inhibit tumor growth via the reduction in M2 polarization of TAMs in mouse models. In vitro evidence reinforced this. Fasting was shown to induce cancer cell autophagy.
Zahra et al. [33]	2017	MIA PaCa-2 injected athymic nu/nu murine models	Pancreatic	*n =* 6 (total), *n =* 3 (standard chow), *n =* 3 (4:1 KetoCal)	Mice injected with MIA PaCa-2 cells were fed 2 different diets: a standard isocaloric diet and a specific KetoCal diet (KD analogue). Mice were then exposed to 6 fractions of 2 Gy for 25 days. Tumor progression was measured.	KD enhanced radiation response in mouse xenograft models without increasing radiation toxicity.
Lo Re et al. [49]	2017	LX-2 human hepatic stellate cell (HSC) line	NA	1 cell line	LX-2 cells were exposed to either NM or FM, and some were then subsequently treated with LPS.	LX-2 cells exposed to FM had inhibited DNA synthesis/proliferation rates. LPS managed to increase LX-2 cell proliferation by, regardless of media.
		C57BL/6J murine models	NA	*n =* 24 (total), *n =* 12 (72 h fast), *n =* 12 (control)	Mice were administered streptozotocin to simulate human diabetes and non-alcoholic steatohepatitis (NASH) in vivo. One group of mice fasted for 72 h, which was followed by 10 days of high fat diet refeeding.	Cyclic fasting did not have beneficial effect in reducing fibrotic burden of NASH in the model.
		HepG2 and Huh-7 cell lines	Hepatoma	2 cell lines	Cytostatic ability of sorafenib was assessed for HepG2 and Huh-7 cells in either starvation-mimicking (72 h fast) or non-fasting conditions.	Fasting synergized with and increased efficacy of the anti-proliferative effects of Sorafenib.
Zhang et al. [50]	2018	Various human cancer cell lines	Glioma, leukemia, pancreatic, thyroid, cervical, liver, renal, lung, gastric, colorectal, prostate, breast, and ovarian	33 cell lines	Expression levels of 4 enzymes were examined: BDH 1/2 and OXCT 1/2.	No correlation between glycolysis reliance and ketolytic enzyme quantity. BHB-supplemented HeLa cells saw improved growth rates; PANC-1 saw no change. BHB no longer benefitted HeLa growth with BDH1/OXCT1 downregulation via siRNA.
		nu/nu murine models	Pancreatic	NA	BDH1/2 and OXCT1/2 levels were investigated in PANC-1 and HeLa xenograft tumors.	In vivo KD results were consistent with in vitro data. PANC-1 xenograft tumor growth rate was heavily attenuated compared to HeLa. Mean survival of KD HeLa mice was significantly lower than that of standard diet HeLa mice.
Manukian et al. [18]	2019	Balb/c murine models	TNBC	NA	Mouse models were injected with 4TI cells and randomized into AL or CR. CR mice were treated with either only CR or CR + RT (12 Gy in 3 fractions).	Increased expression activity related to integrin signaling and inflammation pathways was observed. CD4 + CD25 + Foxp3+ Tregs were significantly decreased with CR. Effector T cells from CR mice produced significantly more interferon-y than AL controls.
Li et al. [51]	2021	C57BL/6J murine models	NA	NA	Mouse models were fed ad libitum or subjected to 30% CR before TBI or TAI treatment.	CR improved intestinal architecture and restored hematogenic organs in all mice. Male mice noted improved gut barrier function; female mice had mitigated systemic and enteric inflammation. There were notable differences in gut microbiome bacterial species enrichment in either sex.

STS: short-term starvation, MMS: methylmethane sulfonate, CP: cyclophosphamide, DSR: differential stress resistance, AL: ad libitum, EtO: etoposide, KD: ketogenic diet, VEGF: vascular endothelial growth factor, BHB: beta-hydroxybutyrate, TMZ: temozolomide, ADF: alternate day feeding, CR: caloric restriction, RES: resveratrol, OXP: oxaliplatin, TAM: tumor-associated macrophages, NM: normal media, FM: fasting media, LPS: lipopolysaccharide, NASH: non-alcoholic steatohepatitis, BDH: 3-hydroxyburyrate dehydrogenase, OXCT: succinyl-CoA:3-oxoacid CoA transferase, BHB: beta-hydroxybutyrate, TBI: total body irradiation, and TAI: total abdominal irradiation dehydrogenase.

**Table 3 cancers-14-05149-t003:** Current Clinical Studies that are currently undergoing or have just completed accrual for examining the effects of dietary intervention (KD, IF, CR) in patients who were undergoing CT and/or RT.

ClinicalTrials.gov Identifier	Responsible Party	Status of Trial	Trial Start Date	Trial Completion Date	Patient Eligibility Criteria	Type(s) of Cancer	Treatment Regimen	Diet	Outcome Measure(s)
NCT05359848	Roberto Pili, State University of New York at Buffalo	Recruiting	15 April 2022	1 November 2023	Adult cancer patients eligible for chemotherapy with adequate hematologic, renal, and liver function with >6 months life expectancy	any	CT	CR/IF	Feasibility and compliance of IF/CR plant-based diet in CT patients
NCT03535701	Jeff Volek, Ohio State University Comprehensive Cancer Center	Completed	20 October 2017	1 May 2020	Adult; BMI ≥ 22; FDG-18 PET avid tumor; life expectancy >6 m; ECOG performance status 0–1	metastatic or stage IV breast cancer	CT	KD	Feasibility and compliance to KD (tumor progression and biologic and behavioral health markers)
NCT03591861	Washington University School of Medicine	Recruiting	1 May 2019	31 May 2034	<21 y/o; life expectancy >12 w; Karnofsky/Lansky performance score ≥60	recurrent primary brain tumor with no curative therapy	CT	KD	Feasibility of combining KD with CT in children with relapsed brain tumors
NCT05234502	Mehmet Artac, Necmettin Erbakan University	Not yet recruiting	30 January 2022	30 January 2025	19–64 y/o; female; Dx with breast cancer 1st time; plan to receive neoadjuvant treatment; BMI 25–29.9 or ≥30; Karnofsky Performance Score ≥70	breast cancer	CT	KD	Interventional; Effects of KD on metabolism and polyneuropathy on patients receiving neoadjuvant CT (polyneuropathy, polyneuropathy severity of symptoms, tumor size, QoL, 5-year survival rate)
NCT04631445	Translational Drug Development	Recruiting	2 December 2020	1 June 2023	adult; Karnofsky Performance Status of ≥70; Life expectancy ≥12 w	metastatic pancreatic ductal adenocarcinoma	CT	KD	Evaluate the progression free survival, changes in serum metabolites and quality of life (KD + triplet therapy)
NCT01535911	Kenneth Schwartz, MD, Michigan State University	Active, not recruiting	1 April 2012	1 June 2024	18–90 y/o; ECOG performance status ≤2; Life expectancy >3 m	GBM	CRT	Energy restricted KD	Effect of KD with RT/CT on tumor size
NCT04730869	Waikato Hospital	Recruiting	26 May 2021	1 November 2022	≥18 y/o; ECOG Performance Status 0–2	GBM	CRT	KD/IF	Feasibility, safety, and efficacy of a Metabolic Therapy Program in Conjunction with Standard Treatment for Glioblastoma Multiforme
NCT05428852	Jeff Volek, Ohio State University Comprehensive Cancer Center	Recruiting	1 July 2022	1 July 2024	18–75 y/o; Graded Prognostic Assessment >1.5; BMI ≥ 18; ECOG performance status 0–1	brain metastasis	RT	KD	Feasibility, metabolic outcome, effect on cognitive function, QOL of KD versus AICR diet on brain metastasis patient selected for radiosurgery or undergoing RT
NCT03451799	Jethro Hu, Cedars-Sinai Medical Center	Active, not recruiting	13 April 2018	1 October 2023	≥18 y/o; Karnofsky Performance Status >70; BMI >22	GBM	CRT	KD	Safety and feasibility of KD in combination with standard-of-care radiation and temozolomide for patients with glioblastoma
NCT02302235	Mid-Atlantic Epilepsy and Sleep Center, LLC	Completed	1 February 2014	1 May 2022	18–65 y/o; Documented surgical resection/debulking; Karnovsky Performance Score ≥70	GBM	CRT	KD	Efficacy, safety, tolerability of KD as adjunctive treatment of RT in GBM
NCT05327608	Thomas Jefferson University	Recruiting	28 July 2022	1 May 2024	≥18 y/o; BMI ≥ 25; HER2-negative; ER/PR <10%; clinical stage I-III; size ≥1.5 cm;	primary invasive breast carcinoma	CT	IF	Feasibility of IF for patients with HER2- negative and ER/PR <10% breast cancer and body mass index ≥ 25 receiving neoadjuvant CT
NCT03162289	Andreas Michalsen, Charite University, Berlin, Germany	Recruiting	10 May 2017	10 June 2022	18–75 y/o; conventional adjuvant or neo-adjuvant protocol with at least 4 CT cycles	non-metastatic ovarian or breast cancer	CT	IF	Tolerance and effectiveness of CT through accompanying intense nutritional therapy interventions
NCT05259410	Kelsey Nicole Dipman Gabel, University of Illinois at Chicago	Not yet recruiting	1 July 2022	1 March 2026	18–99 y/o; ECOG 0 or 1; BMI 25–40	Breast cancer (histologically confirmed Stage I-III)	CT	IF	Feasibility, safety, and efficacy of time-restricted eating combined with CT during breast cancer treatment
NCT02710721	Andreas Michalsen, Charite University, Berlin, Germany	Active, not recruiting	1 April 2016	1 December 2021	25–89 y/o male; BMI ≥ 20	CRPC or hormone-sensitive prostate cancer with high metastatic load	CT	IF	Efficacy of fasting and nutritional therapy as a complementary treatment of advanced metastatic prostate cancer undergoing CT
NCT05503108	J.R. Kroep, Leiden University Medical Center	Not yet recruiting	15 October 2022	15 April 2032	≥18 y/o; WHO performance status 0–2; Clinical stage II-III (cT1cN+ or ≥T2 any cN, cM0); BMI ≥ 18.5	HR+, HER2- Breast Cancer	CT	IF	Response rate, effect, side effects and QoL of IF during neoadjuvant CT
NCT04626843	Eleah Stringer, British Columbia Cancer Agency	Active, not recruiting	3 February 2021	31 December 2022	19–85 y/o; BMI ≥ 20; ECOG Performance Status ≤2; Lymphocytes ≥40 and <150; Hb ≥ 100 g/L	CLL and SLL	CT	IF	Feasibility and efficacy of IF in CLL/SLL Patients
NCT05023967	National Cancer Institute (NCI)	Not yet recruiting	30 January 2023	30 November 2024	≥18 y/o; ECOG performance status ≤ 1 (Karnofsky ≥ 70%); BMI >18.5	Invasive Breast Cancer or DCIS	CT	IF	Combined effect of prolonged nightly fasting and metformin in decreasing breast tumor cell proliferation and other biomarkers of breast cancer
NCT01819233	Thomas Jefferson University (Sidney Kimmel Cancer Center at Thomas Jefferson University)	Completed	8 March 2013	1 September 2019	≥18 y/o; female; BMI ≥ 21; Weight ≥ 100 lbs	DCIS or invasive breast cancer	RT	CR	Feasibility of RT with CR for the treatment of breast cancer; measurable changes of patient characteristics and tissue and serum from CR conditions to determine a metric for evaluating this treatment in future studies
NCT04959474	Thomas Jefferson University	Recruiting	23 August 2021	1 January 2026	≥40 y/o; BMI ≥ 21; KPS score 70–100; tumor size ≤ 3.0 cm; unifocal	DCIS or invasive breast cancer	RT	CR	Compare percent reduction in cellularity of breast tumor between pre-operative SABR with and without CR
NCT01802346	University of Southern California	Recruiting	29 January 2013	29 January 2024	≥19 y/o; BMI ≥ 18.5; ECOG performance status 0–1	breast cancer or metastatic prostate adenocarcinoma	CT	CR	Impact on toxicity and efficacy of CR prior to CT in treatment of breast and prostate cancer
NCT02827370	Thomas Jefferson University (Sidney Kimmel Cancer Center at Thomas Jefferson University)	Active, not recruiting	16 June 2016	1 April 2022	≥18 y/o; female; BMI ≥ 21; Weight ≥ 120 lbs; Karnofsky Performance Status of 80–100%	invasive breast cancer (Non-metastatic and non-inflammatory)	CT	CR	CR impact on pathologic complete response rate, incidence of adverse events and efficacy of CT for breast cancer
NCT03340935	Filippo de Braud, Fondazione IRCCS Istituto Nazionale dei Tumori, Milano	Completed	1 February 2017	31 July 2020	≥18 y/o; BMI ≥ 20	any, except small cell neuroendocrine tumors	standard adjuvant treatments or therapies for advanced disease	CR	Safety, feasibility and metabolic effects of the fasting-mimicking diet (FMD) in cancer patients
NCT05082519	Etan Orgel, Therapeutic Advances in Childhood Leukemia Consortium	Recruiting	12 March 2022	15 October 2031	10–25 y/o; Karnofsky > 50% for patients > 16 y/o and Lansky > 50% for patients ≤16 y/o; BMI% ≥ 5th percentile for age for patients aged 10–19 years, BMI ≥ 18.5 in patients 20–29 years	B-ALL	CT	CR	Efficacy of the IDEAL2 (Improving Diet and Exercise in ALL) CR and activity intervention integrated into HR B-ALL induction to reduce incidence of end of induction MRD ≥ 0.01%; efficacy of the IDEAL2 intervention to reduce gain in fat mass during induction

CT: chemotherapy, IF: intermittent fasting, CR: caloric restriction, KD: ketogenic diet, QoL: quality of life, CRT: chemoradiation therapy, GBM: glioblastoma, DCIS: ductal carcinoma in situ, CLL: chronic lymphocytic leukemia, SLL: small lymphocytic lymphoma, B-ALL: B-cell acute lymphoblastic leukemia.

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
