# Peer review of "Dietary Interventions in Cancer Treatment and Response: A Comprehensive Review"

_cancers, 2022, doi:10.3390/cancers14205149_

Round 1

Reviewer 1 Report

This work presents in a non-exhaustive way the pre-clinical and clinical data concerning caloric and carbohydrate restriction diets during cancer treatments.
I find that the methodological criticism of the cited studies is not sufficiently supported and objective. One cannot draw conclusions leading to recommendations with such a low level of evidence, no matter how much belief one may have in these diets.
I remind you that the ESPEN in these latest guidelines concerning patients with cancer, recommends not using these diets in patients at risk of malnutrition, ie in nearly 60% of patients followed in oncology.
I invite the authors to review their analysis of the clinical data and their conclusions with a more objective view of the current state of knowledge on this subject.

Author Response

Point 1: This work presents in a non-exhaustive way the pre-clinical and clinical data concerning caloric and carbohydrate restriction diets during cancer treatments.
I find that the methodological criticism of the cited studies is not sufficiently supported and objective. One cannot draw conclusions leading to recommendations with such a low level of evidence, no matter how much belief one may have in these diets.

 Response 1:

We appreciate all the comments this review has provided. For the reviewer’s first point, we would like to comment that this paper is a comprehensive review on the role of dietary interventions in both preclinical and clinical studies. There remains limited clinical investigations on this subject. This is one of our main conclusions and the basis for our advocacy in expanding upon what little published research currently exists.

We agree with the reviewer that some of our claims about these dietary interventions’ potential in improving treatment outcome and patient Quality of Life to be a bit too subjective since there is not a lot of data concerning the topic. We have since modified our conclusions pertaining to said studies and included additional text to specify that existing data is insufficient to guide clinical practice.

Point 2: I remind you that the ESPEN in these latest guidelines concerning patients with cancer, recommends not using these diets in patients at risk of malnutrition, ie in nearly 60% of patients followed in oncology.

Response 2:

We appreciate the reviewer's comment, but we were not familiar with the “ESPEN guidelines” at the time of writing. We appreciate that the reviewer pointed out that some cancer patients are at risk of malnutrition either due to their cancer or due to toxicity from treatment. Thus, we have further emphasized that the potential implementation of these diets should be taken with great caution and under the monitoring of a physician and dietician in the context of a prospective randomized clinical trial. We have also included additional statements discussing. the potential risks of implementing such diets in a population that is vulnerable to malnutrition, according to the 2021 ESPEN Practical Guideline on clinical nutrition in cancer.

Point 3: I invite the authors to review their analysis of the clinical data and their conclusions with a more objective view of the current state of knowledge on this subject.

Response 3:

We appreciate and fully acknowledge the reviewer’s last comment. As such, we have adjusted our conclusions and implemented several of the elements of the dietary interventions requested by the reviewer. We have now emphasized the limitations of existing studies to present our findings in a more objective and comprehensive light.

Reviewer 2 Report

This is a nice overview of the current state of knowledge on dietary interventions in cancer treatment. The review nicely includes several important studies in the field. There are a few areas in which the language/grammar could be tighter, but I believe this article will be of interest to many in the field of cancer.

1.       Line 108: the authors state that – “Preliminary studies performed in animal models have shown 108 promise in increasing the beneficial effects of RT for the treatment of cancers in transgenic 109 and transplant mouse models of neuroblastoma, fibrosarcoma, glioma, melanoma, breast, 110 and ovarian cancers [18].” The authors need to correct this sentence to support the section and the reference.

2.       In figure 1, the authors should describe QoL. Quality of life. As done for the tables.

3.       Figure 1 can be expanded to show how IF, CT, and KD interact with the different treatment modalities to result in different prognostic outcomes.

4.       Line 133: the authors state that – “Table 2. Pre-Clinical studies examining the effect of dietary interventions (CR, IF, KD) in patients undergoing Chemotherapy and/or Radiotherapy.” Yet, the table does NOT involve any human patients. Studies are in mice. Please correct.

5.     While given that the authors have provided past work which may or may not have prognostic benefits relating to diet, it is clear that several studies are currently underway wherein dietary interventions are being probed. The authors should offer various dietary interventions that can be potentially paired with pharmacological drugs as it is unlikely that diet alone will be strong enough against tumor growth, for future study (or currently recruiting). Combinations for CR, IF, and KD. For example:

https://clinicaltrials.gov/ct2/show/NCT04986670

6.     The authors need to add more clarity to figure 2 as it lacks enough information on its own.

Author Response

This is a nice overview of the current state of knowledge on dietary interventions in cancer treatment. The review nicely includes several important studies in the field. There are a few areas in which the language/grammar could be tighter, but I believe this article will be of interest to many in the field of cancer.

We appreciate the reviewer's feedback, and we made adjustments throughout the manuscript in response.

 Point 1: Line 108: the authors state that – “Preliminary studies performed in animal models have shown 108 promise in increasing the beneficial effects of RT for the treatment of cancers in transgenic 109 and transplant mouse models of neuroblastoma, fibrosarcoma, glioma, melanoma, breast, 110 and ovarian cancers [18].” The authors need to correct this sentence to support the section and the reference.

Response 1:

With this reviewer’s first point on line 108, we have corrected the source cited to support our section and the claim that was being made.

Point 2:   In figure 1, the authors should describe QoL. Quality of life. As done for the tables.

Response 2:

In figure 1, we have added the definition of Quality of Life (QoL) in the figure legend.

Point 3:  Figure 1 can be expanded to show how IF, CT, and KD interact with the different treatment modalities to result in different prognostic outcomes.

Response 3:

We expanded figure 2 to include the other dietary interventions. The two treatment modalities that we examined, radiotherapy and chemotherapy, both have similar mechanisms of killing cancer cells through direct DNA damage which is described within the context of the figures provided.

Point 4:  Line 133: the authors state that – “Table 2. Pre-Clinical studies examining the effect of dietary interventions (CR, IF, KD) in patients undergoing Chemotherapy and/or Radiotherapy.” Yet, the table does NOT involve any human patients. Studies are in mice. Please correct.

Response 4:

For the title of Table 2, we have included the proper correction.

Point 5: While given that the authors have provided past work which may or may not have prognostic benefits relating to diet, it is clear that several studies are currently underway wherein dietary interventions are being probed. The authors should offer various dietary interventions that can be potentially paired with pharmacological drugs as it is unlikely that diet alone will be strong enough against tumor growth, for future study (or currently recruiting). Combinations for CR, IF, and KD. For example:

https://clinicaltrials.gov/ct2/show/NCT04986670

Response 5:

We have included a new table detailing the ongoing clinical trials concerning varying types of cancer that implement the different dietary interventions described in this paper.

 Point 6: The authors need to add more clarity to figure 2 as it lacks enough information on its own.

Response 6:

We have expanded upon the description of figure 2 to better explain and clarify the purpose of the figure.

Reviewer 3 Report

The manuscript "Dietary Interventions in Cancer Treatment and Response: A Comprehensive Review" adds knowledge to the field and presents potentially interesting findings. Nevertheless, some questions should be addressed in order to improve its scientific quality:

- The review is generally well written and clear but contains a number of typos and grammatical errors - these should be corrected throughout.

- Titles of each subsection should be substitute by a sentence stating the results obtained.

- A concluding sentence is needed at the end of each subsection.

- The Introduction provides sufficient information to understand the state-of-the-art and citations are appropiate.

- Some Figures are too small.

Round 2

Reviewer 2 Report

Point 3:  Figure 1 can be expanded to show how IF, CT, and KD interact with the different treatment modalities to result in different prognostic outcomes.

Response 3:

We expanded figure 2 to include the other dietary interventions. The two treatment modalities that we examined, radiotherapy and chemotherapy, both have similar mechanisms of killing cancer cells through direct DNA damage which is described within the context of the figures provided.

Q. Point 3: There is no expansion in figure 2 which includes “other dietary restrictions”. There is no difference in the previous version of figure 2 and the new version.

Point 5: While given that the authors have provided past work which may or may not have prognostic benefits relating to diet, it is clear that several studies are currently underway wherein dietary interventions are being probed. The authors should offer various dietary interventions that can be potentially paired with pharmacological drugs as it is unlikely that diet alone will be strong enough against tumor growth, for future study (or currently recruiting). Combinations for CR, IF, and KD. For example:

https://clinicaltrials.gov/ct2/show/NCT04986670

Response 5:

We have included a new table detailing the ongoing clinical trials concerning varying types of cancer that implement the different dietary interventions described in this paper.

Q. Point 5. Where is the “new table” detailing the ongoing clinical trials concerning varying types of cancer that implement the different dietary interventions described in this paper?

Author Response

Response to Reviewer 2 Comments

We would like to thank the reviewer for their swift and concise response.

  1. Point 3: There is no expansion in figure 2 which includes “other dietary restrictions”. There is no difference in the previous version of figure 2 and the new version.

Response:

In Q. Point 3 perhaps the reviewer was referring to figure 1 which was previously mentioned in Point 3 of the reviewer’s prior commentary, rather than figure 2?

To address the reviewer’s comments regarding improving the figure, we opted to expand figure 2 in order show how KD, CR, and IF would interact with our chosen treatment modalities (leaving figure 1 as is). The figure 2 of our first submitted draft only included a basic mechanism describing how oxidative stress is affected under IF conditions. Our revised version includes a model for the interactions for KD and CR with the treatment modalities. We have ensured that the revised version of figure 2 is available in the present version of the manuscript for the reviewer to assess.

Previously the reviewer had suggested that we make alterations to figure 1, however we would like to clarify that we instead applied this reviewer’s suggestions to figure 2 to fit his or her recommendations. We felt an expansion to figure 2 (rather than figure 1) was more fitting in meeting the reviewer’s recommendation regarding showing how IF, KD, and CR each interact with the selected treatment modalities.

  1. Point 5. Where is the “new table” detailing the ongoing clinical trials concerning varying types of cancer that implement the different dietary interventions described in this paper?

Response:

Our apologies. We had included the new table (table 3) as part of the Excel file attachment in the ZIP file for our resubmission. As the previous 2 tables had been inserted into the manuscript document for us, we had assumed this new one would be done for us as well. We have inserted it into our new draft of the manuscript ourselves to prevent any further confusion.

Round 3

Reviewer 2 Report

Point 5: While given that the authors have provided past work which may or may not have prognostic benefits relating to diet, it is clear that several studies are currently underway wherein dietary interventions are being probed. The authors should offer various dietary interventions that can be potentially paired with pharmacological drugs as it is unlikely that diet alone will be strong enough against tumor growth, for future study (or currently recruiting). Combinations for CR, IF, and KD. For example:

https://clinicaltrials.gov/ct2/show/NCT04986670

Response 5:

We have included a new table detailing the ongoing clinical trials concerning varying types of cancer that implement the different dietary interventions described in this paper.

Q. Point 5. Where is the “new table” detailing the ongoing clinical trials concerning varying types of cancer that implement the different dietary interventions described in this paper?

  1. Point 5. Where is the “new table” detailing the ongoing clinical trials concerning varying types of cancer that implement the different dietary interventions described in this paper?

Response:

Our apologies. We had included the new table (table 3) as part of the Excel file attachment in the ZIP file for our resubmission. As the previous 2 tables had been inserted into the manuscript document for us, we had assumed this new one would be done for us as well. We have inserted it into our new draft of the manuscript ourselves to prevent any further confusion.

Q. Point 5. The authors must provide title/headings for each of the columns in Table 3. This is currently lacking.

Author Response

Q. Point 5:

We would like to thank the reviewer once more for their concise and straightforward response. In our revised version, we have provided the appropriate column headings for Table 3.